Metacarpal torsion in apes, humans, and early Australopithecus: implications for manipulatory abilities

Drapeau Michelle S.M. m.drapeau@umontreal.ca
Département d’Anthropologie, Université de Montréal , Montréal , Canada
Reno Philip
Electronic publication date: 2015 Oct 6
Publication date: 2015
Volume: 3
Electronic Location ID: e1311
Received 2015 May 29; Accepted 2015 Sep 19
Copyright: © 2015 Drapeau
Copyright year: 2015
Copyright holder: Drapeau
License: This is an open access article distributed under the terms of the Creative Commons Attribution License, which permits unrestricted use, distribution, reproduction and adaptation in any medium and for any purpose provided that it is properly attributed. For attribution, the original author(s), title, publication source (PeerJ) and either DOI or URL of the article must be cited.
License URL: https://creativecommons.org/licenses/by/4.0/

Keywords: Metacarpal, Torsion, Australopithecus, Human, Hominoid, Manipulation, Hominin, A. afarensis, A. africanus, Swartkrans

Funding: Fond Québecois de la Recherche sur la société et la culture 2006-NP-108312 This study was funded in part by the Fond Québecois de la Recherche sur la société et la culture (2006-NP-108312). The funders had no role in study design, data collection and analysis, decision to publish, or preparation of the manuscript.

==============================
Human hands, when compared to that of apes, have a series of adaptations to facilitate manipulation. Numerous studies have shown that Australopithecus afarensis and Au. africanus display some of these adaptations, such as a longer thumb relative to the other fingers, asymmetric heads on the second and fifth metacarpals, and orientation of the second metacarpal joints with the trapezium and capitate away from the sagittal plane, while lacking others such as a very mobile fifth metacarpal, a styloid process on the third, and a flatter metacarpo-trapezium articulation, suggesting some adaptation to manipulation but more limited than in humans. This paper explores variation in metacarpal torsion, a trait said to enhance manipulation, in humans, apes, early australopithecines and specimens from Swartkrans. This study shows that humans are different from large apes in torsion of the third and fourth metacarpals. Humans are also characterized by wedge-shaped bases of the third and fourth metacarpals, making the metacarpal-base row very arched mediolaterally and placing the ulnar-most metacarpals in a position that facilitate opposition to the thumb in power or cradle grips. The third and fourth metacarpals of Au. afarensis are very human-like, suggesting that the medial palm was already well adapted for these kinds of grips in that taxon. Au. africanus present a less clear human-like morphology, suggesting, perhaps, that the medial palm was less suited to human-like manipulation in that taxa than in Au. afarensis. Overall, this study supports previous studies on Au. afarensis and Au. africanus that these taxa had derived hand morphology with some adaptation to human-like power and precision grips and support the hypothesis that dexterous hands largely predated Homo.

Introduction

Much of the debate on Australopithecus has focused on its locomotor habits and the maintenance (or not) of an arboreal component. However, manipulatory capabilities in that taxon have also been argued (e.g., Marzke, 1983; Marzke, 1997; Susman, 1998; Drapeau, 2012; Kivell et al., 2011; Skinner et al., 2015). Marzke (1997) and Marzke (2005) identified three traits that suggest that the hand of one of the oldest Australopithecus species, Au. afarensis had hands that were able to produce better precision grips and handling than the ape hand: a more robust and longer thumb relative to the other fingers, asymmetric heads on the second and fifth metacarpals, and orientation of the second metacarpal (MC) joints with the trapezium and capitate away from the sagittal plane. However, Susman (1998) doubts that all these traits indicate significant improvement of manipulatory skills. Interestingly, they both recognize that the radial torsion (toward the thumb) of the second and third MC heads improves manipulatory grips (Susman, 1979; Marzke & Shackley, 1986; Marzke, 1997; Marzke, 2005), although neither recognizes that trait in Australopithecus (but see Kivell et al., 2011; Supplemental Information). Torsion of the second and third MC head in hominoids is a trait that is described qualitatively, but has not been extensively quantified (except in humans; Singh, 1979; Peters & Koebke, 1990) and rarely statistically compared among humans and great apes (Drapeau, 2009). This paper explores MC head torsion in these extant species and compares values for Au. afarensis, Au. africanus and Swartkrans specimens.

Background

Humans and apes have different power grips. Humans hold objects obliquely in the cupped palm, positioning the thumb in opposition to the other fingers. The fifth digit is the most flexed and the subsequent lateral fingers, digits 4 to 2, are progressively less flexed (Lewis, 1977; Lewis, 1989; Napier, 1993; Kapandji, 2005). The two ulnar-most MCs are also slightly flexed at the carpometacarpal joint. In addition, the palmar surfaces of the fingers are supinated, i.e., turned toward the thumb. Apes, instead, flex digits 2 to 5 in a hook-like fashion, with no obvious differences in degree of flexion among the fingers and no apparent flexion at the carpometacarpal joint (Napier, 1960; Lewis, 1977; Lewis, 1989).

However, the hand is not used solely for powerful gripping, but is important for fine manipulation as well, particularly in humans. In precision grips, the thumb is opposed to the other fingers and objects are often held with the finger pads or palmar surface of the fingers (Napier, 1956). Depending on the size of the object held, the position of the ulnar digits varies. When manipulating small objects in a pad-to-pad grip, only the second or second and third digits are opposed to the thumb (Napier, 1956; Marzke & Shackley, 1986). In that position, the second and third digits are supinated. However, as the size of the object held increases or if the grip involves the palmar surface of the digits rather than the pads, the third and particularly the second digit tend to take a more pronated position (Napier, 1956). In addition, the fourth and fifth digits become involved and take a supinated position (Marzke & Shackley, 1986).

Apes are much less dexterous than humans in manipulation (Napier, 1960; Napier, 1962) and have much more difficulty in pad-to-pad grips (Christel, 1993). This is in part a consequence of their relatively long fingers and short thumbs (e.g., Mivart, 1867; Schultz, 1930; Ashley-Montagu, 1931; Green & Gordon, 2008). They are able to handle small objects between the thumb and the side of the phalanges of the index or between the tips of the thumb and fingers (Napier, 1960; Napier, 1962; Christel, 1993). This particular grip does not require marked rotation of the second digit. In contrast to humans, the morphology of the ape hand is most likely strongly driven by locomotor constraints. All great apes are characterized by a hook grip, which involves flexing all the fingers in sagittal planes (Lewis, 1977; Lewis, 1989; Napier, 1960; Napier, 1993).

Morphological adaptations to the different grips observed in humans can be seen in their hands. First, as mentioned above, the thumb is robust and digits 2–5 are much shorter relative to the thumb in humans than in apes, and also relative to measures of body size (Mivart, 1867; Drapeau & Ward, 2007; Lovejoy et al., 2009; Supplemental Information; Almécija, Smaers & Jungers, 2015). The third MC base has a styloid process that is hypothesized to resist palmar dislocation of the base (Marzke, 1983). Also, the bases of the fourth and fifth MCs allow for some axial motion (El-shennawy et al., 2001). The base of the second MC has a saddle shape joint with the trapezoid rather than the wedge shape observed in apes (Marzke, 1983). The base of the MC2 has a continuous articulation with the capitate instead of an articulation that is bisected in anterior and posterior segments by carpometacarpal ligaments as found in all extant apes (Lewis, 1973; Lewis, 1977; Lewis, 1989; Marzke, 1983; Tocheri et al., 2008; Drapeau, 2012). In apes, that joint is in a sagittal plane while in humans it is oriented more transversely. That articulation and the articulation between the second and third MCs are anteroposteriorly curved instead of being flat as in apes (Lewis, 1973; Lewis, 1977; Lewis, 1989; Marzke, 1983; Tocheri et al., 2008; Drapeau, 2012). Finally, the second MC-trapezium articulation lies in a more transverse plane instead of the sagittal plane found in apes (Marzke, 1983; Drapeau et al., 2005; Tocheri et al., 2008). Although no significant pronation-supination has been recorded in human cadavers at that joint, its morphology allows for some mobility in flexion-extension (Batmanabane & Malathi, 1985; El-shennawy et al., 2001). Similarly, the base of the human fifth MC is saddle shaped with a dorsoventral convexity. This morphology, combined with a retraction of the hook of the hamate, allows for flexion and supination of the MC (Dubousset, 1981; Marzke, 1983; Lewis, 1989; El-shennawy et al., 2001; Reece, 2005; Buffi, Crisco & Murray, 2013). Although the base of the fourth MC is not as clearly saddled-shape, it is also quite mobile in flexion-extension and in pronation-supination (El-shennawy et al., 2001). In great apes, the ventral surface of the base articulates with the hook of the hamate, limiting flexion and axial rotation (Lewis, 1989).

The head shape of MCs 2 and 5 is also modified to allow rotation of the fingers. The MC2 head has a distal articular surface whose palmo-radial corner projects more proximally (Lewis, 1989; Drapeau, 2012). In palmar view, the head is slanted radially (Lewis, 1989; Drapeau, 2012). This morphology, combined with the collateral ligaments, causes the proximal phalanx to deviate radially and to pronate when flexed (see Lewis, 1989 for details). The extended finger tends to be supinated when ulnarly deviated and pronated when radially deviated or flexed. The morphology of the third MC head also presents some asymmetry, but to a lesser degree than the second and the rotation and axial deviation of the phalanx are also less marked than in the second (Lewis, 1989). The morphology of the fifth MC head is the mirror image of the second, causing movements in opposite directions relative to the second MC (Lewis, 1989). This particular morphology of the heads, combined with the morphology of the bases, allows for axial rotation of the second and fifth fingers and MCs. In contrast, the ape’s MCs do not have such asymmetrical heads and movement at these joints function much more as simple hinges (Lewis, 1989; Drapeau, 2012). The human first MC allows for a greater range of thumb opposition because the base has a less projecting palmar beak than what is seen in chimpanzees and the articulation is flatter in the dorso-palmar direction (Marzke, 1992; Tocheri et al., 2003). The morphology of the head, with a palmar elevation radially, allows for some pronation and abduction of the distal segments of the thumb during flexion, but it is a morphology that appears to be primitive in hominoids and even in mammals (Lewis, 1989).

Australopithecines have some of the traits associated with manipulation in humans. For example, Au. afarensis is characterized by a continuous and curved MC2-capitate articulation that is more transversely oriented than in apes, but less than in humans (Marzke, 1983; Ward et al., 1999; Drapeau et al., 2005; Ward et al., 2012), an MC2-trapezium facet that is more transversally oriented than in chimpanzees (Marzke, 1983; Drapeau et al., 2005; Ward et al., 2012), asymmetric MC2, MC3 heads (Susman, 1979; Bush et al., 1982; Marzke, 1997; Ward et al., 1999; Drapeau, 2012; Ward et al., 2012), long thumbs relative to the other fingers (Alba, Moyà-Solà & Köhler, 2003; Almécija & Alba, 2014; but see Rolian & Gordon, 2013 for opposing view), and short fingers 2-5 relative to measures of body size (Drapeau et al., 2005; Drapeau & Ward, 2007), traits that are also found in Au. africanus (Ricklan, 1990; Clarke, 1999; Green & Gordon, 2008; Tocheri et al., 2008; Kivell et al., 2011; M Drapeau, pers. obs., 2011). These traits suggest the capacity to make a 3-finger chuck. However, a volar beak on the MC1 possibly restricted opposition of the thumb (Marzke, 1992; but see Ricklan, 1990 for an opposing view). In addition, Au. afarensis had slightly asymmetric MC5 heads (Ward et al., 2012). However, this taxa and Au. africanus, because of a palmar articulation with the hook of the hamate, were probably more limited than humans in MC5 flexion and supination (Marzke, 1983; but see Ricklan, 1987 for an opposing view). These traits suggest that the fossils may have had some mobility of the medial most carpometacarpal joint, but could not produce completely human-like power and cradle grips with a cupped palm. Comparable to Au. afarensis and Au. africanus, the hand of more recent Au. sediba presents asymmetric MC heads, a continuous and curved, proximolaterally facing facet between the capitate and MC2, a long thumb relative to the other fingers, and short ulnar fingers relative to a measure of body size (Kivell et al., 2011). Unfortunately, the morphology of these traits is unknown for the Swatkrans specimens. Little is known of the hand of Au. anamensis, but a fairly abraded capitate suggest that it had a discontinuous capitate-MC2 facet (Ward, Leakey & Walker, 2001) oriented at 90° from the MC3 articular facet like apes and unlike the more transverse orientation of humans and, to a lesser degree, Au. afarensis (Leakey et al., 1998; Ward, Walker & Leakey, 1999; Ward, Leakey & Walker, 2001). Ardipithecus ramidus, dated at 4.4 Ma, also presents some human-like traits: a continuous MC2-capitate surface as well as a mobile hamate-MC5 joint, a long thumb relative to the fingers and short fingers 2–5 relative to measures of body size (Lovejoy et al., 2009).

With the exception of Au. anamensis, the hands of hominin taxa display human-like traits that suggest that they were generally capable of manipulations with three-finger chuck and pad-to-pad grips (see also Almécija, Moyà-Solà & Alba, 2010 for Orrorin tugenensis). In this study, I contrast MC head torsion in human and great apes to show how it reflects the differences in grips between extant taxa. I also compare Au. afarensis, Au. africanus and specimens from Swartkrans to extant species to evaluate their morphological affinities and possibly identify additional traits related to manipulation in the fossil specimens.

In hominoids, the bases of the MCs are disposed in a mediolateral arch configuration (Fig. 1), with the concave, palmar side housing the carpal tunnel (although much of the walls of the tunnel are the result of the projecting hook of the hamate and of the position of the trapezium usually disposed at an angle from the other distal carpal bones; see Lewis, 1989) and Reece (2005) observed that humans had more arched rows than apes. Metacarpals are expected to present torsion values that adjust for the degree of arching. As a result, the ulnar-most digits will tend to have heads that are more ulnarly twisted, while the radial-most digits (except the thumb) will tend to have heads that are more radially twisted.

Figure 1 Palmar arch.

Metacarpals two to five of a left hand in distal view illustrating the arch formed by the metacarpal bases (modified from Peters & Koebke, 1990).

More specifically, humans, because of the types of grips described above, are expected to have, on average, MC 2–5 heads that are more radially twisted than apes. However, variation in arching of the MC row is expected to influence the twisting of the MCs. For example, ulnar digits may not present as much torsional difference as the more radial digits in a hand that would have greater arching. In addition, because base and head morphologies of the second MC and particularly of the fifth MC of humans allow for axial rotation of the digit to conform to various object sizes and shapes, torsion of these two MCs may not be as different from apes as for the other digits. In humans, the trapezoid is wider palmarly than that of apes, which pushes the trapezium radially and rotates it into alignment relative to the rest of the proximal carpal row (Tuttle, 1970; Lewis, 1977; Lewis, 1989; Sarmiento, 1994; Drapeau et al., 2005; Tocheri et al., 2005). As a result, the trapezio-MC articulation lies within an axis comparable to that of the other digits. This reorientation is accompanied by a palmar expansion of the articular facet between the trapezoid and capitate. Together, these traits (palmar expansion of the trapezoid, a first carpal metacarpal articulation in line with the rest of the carpal row, and an expanded palmar trapezoid-capitatum articulation) allow for large loads incurred at the base of the MC1 during forceful grips to be dissipated through the palmar carpal row (Lewis, 1977; Lewis, 1989; Tocheri et al., 2005). Because of the reorientation of the first carpometacarpal joint in humans, a greater torsion of the first MC is expected in order to bring the metacarpo-phalangeal joint in an axis perpendicular to that of the other digits. Apes, particularly chimpanzees, having the base of the first MC already perpendicularly rotated relative to the other carpometacapal joints, are not expected to require as much torsion of the first MC to function in opposition to the other digits or to the palm.

Materials

The human sample is from archaeological collections and it consists of a mix of Euroamericans from the 19th century and of Canadian Amerinds (Table 1). All extant great apes are wild shot and the Gorilla sample includes only western lowland gorillas. All specimens are free of pathologies. Sample size varies from one MC to the other as a function of the availability of each bone in the osteological collection (Table 2).

Table 1 Comparative sample for torsion values.

Species	Male	Female	Indet.	Total	
Homo sapiens (total)	20	11	17	48	
Euroamericans	8	1	5	14	
Amerinds	12	10	12	34	
Pan troglodytes	16	25		41	
Gorilla gorilla	27	20		47	
Pongo pygmaeus	13	17		30	

Table 2 Torsion values for the extant taxa.

Extent species descriptive statisticsa for torsion anglesb.

Taxon	MC1	MC2	MC3	MC4	MC5	
H. sapiens	6.5	−14.0	−21.2	9.6	10.9	
	8.1	7.2	6.8	7.6	7.0	
	43	46	43	42	38	
P. troglodytes	−16.7	−12.9	−6.5	2.4	5.5	
	5.7	6.7	6.3	7.1	9.1	
	27	39	40	40	39	
G. gorilla	−7.9	−11.5	−9.4	2.7	10.1	
	8.9	5.8	7.7	5.7	8.7	
	39	42	44	44	44	
P. pygmaeus	10.8	−18.6	−9.8	3.5	6.2	
	10.6	8.8	8.2	6.4	6.0	
	29	29	29	29	29	
Notes.

a The mean is presented on the first line, standard deviation on the second, and sample size on the third.

b In degrees. Positive values represent heads with their palmar side that are twisted ulnarly relative to the base (away from the thumb), negative values represent heads twisted radially (turned towards the thumb).

The hominin fossils included in this analysis are from Hadar, Ethiopia, and from Sterkfontein and Swartkrans, South Africa (Table 3). Specimens from Hadar are all attributed to Au. afarensis (Bush et al., 1982; Drapeau et al., 2005; Ward et al., 2012) and date at 3.2 Ma for A.L. 333 and 3 Ma for A.L. 438 (Kimbel, Rak & Johanson, 2004). Although some have argued that there might be more than one species represented at Sterkfontein (e.g., Clarke , 2013), all three Sterkfontein specimens included are from Member 4 and are assumed to belong to Au. africanus based on the general morphology, size and provenience (McHenry & Berger, 1998), and date between 2.6 and 2 Ma (Herries et al., 2013). At Swartkrans, Paranthropus robustus and early Homo are present and post-cranial specimens are difficult to assign to either of these taxa with certainty. SK 84 is from Member 1 and attributed to Homo (Susman, 1994; Susman, 2004), SKW 2954 is from member 2 and is described as being human-like (Susman, 2004), and SKW 14147 is not assigned to a member or to a specific taxon (Day & Scheuer, 1973). Member 1 is dated between 2.2 and 1.8 Ma and Member 2 between 1.8 and 1 Ma (Gibbon et al., 2014). Only specimens that are complete and undistorted are included in the analysis.

Table 3 Torsion values for the fossils.

Australopithecus afarensis, Au. africanus and Sterkfontein fossil specimens and their torsion values.

Fossil	Element	Side	Torsion angle	
A.L. 333w-39	MC1	R	−14.3	
A.L. 333-48	MC2	L	−1.3	
A.L. 438-1e	MC2	L	−15.0	
A.L. 438-1f	MC2	R	−17.5	
A.L. 438-1d	MC3	L	−22.9	
A.L. 333-16	MC3	L	−23.3	
A.L. 333-56	MC4	L	13.3	
A.L. 333-14	MC5	R	−0.3	
A.L. 333-89	MC5	L	10.5	
A.L. 333-141	MC5	R	−4.0	
Stw418	MC1	L	−10.8	
Stw382	MC2	L	−8.5	
Stw68	MC3	R	−11.8	
SK84	MC1	L	−5.2	
SKW2954a	MC4	R	3.5	
SKW14147	MC5	L	4.0	
Notes.

a Possible healed fracture.

Methods

Using a Microscribe 3DX portable digitizer with a precision of 0.23 mm, palmodorsal axes of the base and head of MCs one through five were recorded to measure head torsion. It was the axis of the whole head that was recorded, irrespective of the asymmetry of the articular surface (Fig. 2). For the MC2, the palmodorsal axis of the base was determined as the margin of the articular surface with the capitate, and for the MC3, it was determined as the margin of the articular surface with the second MC (Fig. 2C). The three-dimensional points were realigned with the software GRF-ND (Slice, 1992–1994) so that x, y, and z values varied in the dorsoplantar, proximodistal and radioulnar anatomical axes respectively. The angle between the lines defining the orientation of the head and of the base in the transverse plane represents the angle of torsion of the MCs. Values presented are for the left hand, but if the measure was not available for one specimen, values from the right were used. Positive values represent heads with their palmar side that are twisted ulnarly relative to the base (away from the thumb), negative values represent heads twisted radially (turned towards the thumb), and a value of zero indicates no torsion relative to the base. In order to estimate the shape of the arch made by the base of the MCs when articulated together, the wedging of the base was measured. It was calculated as the ratio of dorsal width relative to the palmar width of the bases of the MC3 and MC4, the two ‘central’ bones of the arch composed of the four ulnar MCs.

Figure 2 Metacarpal data collection.

Distal (A) and palmar (B) view of human left MC heads, and proximal (with dorsal down) view of the bases (C). The gray points show how the palmodorsal axis of the head and base were recorded with a 3D digitizer (see text for details).

Intraobserver error in angle measurement was estimated with three specimens: Homo, Pan, and Pongo. All five MCs for each specimen were digitized 10 times over a two-day period. Each metatarsal was digitized five times the first day. The second day, the metatarsals were repositioned and recorded another five times. The mean interval of confidence of measurement is ±1.6° and the average range 8.3° (varied from 2.4° to 15.4°). The error was, on average, about twice as high on the pollical MC compared to the others (mean pollical standard error 2.5° vs. 1.3° for the other MCs; mean pollical range 13.5° vs. 7.0° for all other MCs). This error for the first MC is probably due to the fairly round profile of the base (Fig. 2C), which makes the definition of the dorsopalmar axis more difficult to define accurately.

Species are compared with one-way ANOVA and Post hoc multiple comparisons with Bonferroni adjustments when variances are homogeneous among groups and Tamhane T2 tests when heterogeneous.

Results

For the MC1, Homo and Pongo have heads whose palmar surfaces are the most turned towards the other fingers, while Pan has the head that is the least turned towards the other fingers (Table 2). Gorillas are intermediate between these two groups. Homo and Pongo are statistically different from all other extant taxa but are not different from each other (Tables 4 and 5). Australopithecus afarensis (n = 1) is most similar to Gorilla but within the range of all taxa and outside the range of only humans. Australopithecus africanus (n = 1) and the Swartkrans specimen (SK 84) are most similar to Gorilla, but within the range of all species (Fig. 3A).

Figure 3 Boxplot of metacarpal torsion.

Boxplot of the torsion of MC1 to MC5. The box represent the 25–75 quartiles, the horizontal line the median, the whiskers the range, and open and close circles represent outliers and extreme outliers (more than 1.5 and 3.0 standard deviation from the mean).

Table 4 ANOVA-MC torsion.

Results for the one-way ANOVA comparing MC torsion.

Metacarpal	F	Significance	
MC1	68.1	<0.001	
MC2	6.0	0.001	
MC3	33.8	<0.001	
MC4	10.6	<0.001	
MC5	4.4	0.005	

Table 5 Extant taxa comparisons of torsion values.

Post hoc comparisons of torsion values with Bonferroni adjustmenta.

Metacarpal	Taxa	H. sapiens	P. troglodytes	G. gorilla	P. pygmaeus	
MC1	Homo sapiens		23.2	14.4	−4.3	
Pan troglodytes	<0.001		−8.8	−27.5	
Gorilla gorilla	<0.001	<0.001		−18.7	
Pongo pygmaeus	0.2	<0.001	<0.001		
MC2	Homo sapiens		−1.1	−2.4	4.6	
Pan troglodytes	1		−1.3	5.7	
Gorilla gorilla	0.7	1		7.1	
Pongo pygmaeus	0.04	0.007	<0.001		
MC3	Homo sapiens		−14.7	−11.7	−11.4	
Pan troglodytes	<0.001		2.9	3.3	
Gorilla gorilla	<0.001	0.4		0.3	
Pongo pygmaeus	<0.001	0.4	1		
MC4	Homo sapiens		7.2	7.0	6.2	
Pan troglodytes	<0.001		−0.2	−1.0	
Gorilla gorilla	<0.001	1		−0.8	
Pongo pygmaeus	0.001	1	1		
MC5	Homo sapiens		5.4	0.8	4.7	
Pan troglodytes	0.02		−4.6	−0.7	
Gorilla gorilla	1	0.06		3.9	
Pongo pygmaeus	0.1	1	0.2		
Notes.

a Values above the diagonal are absolute mean differences of the pair-wise comparison (row–column), values below are significance of the test (values at 0.05 or less are in bold).

For the MC2, as expected, all species are similarly radially turned towards the thumb except for Pongo that has a significantly more turned MC than the other taxa (Tables 2, 4 and 5). Australopithecus afarensis (n = 2) is variable and does not resemble one taxon in particular. Australopithecus africanus (n = 1) is within the distribution of all taxa, but most similar to African apes (Fig. 3B).

For the MC3, humans have the heads that are the most supinated (Table 2) and are statistically different, while all apes are not significantly different from each other (Tables 4 and 5). Australopithecus afarensis (n = 2) is most similar to humans, while Au. africanus (n = 1) is within the range of all taxa, but most similar to apes (Fig. 3C).

For the MC4, again, humans are statistically different from all apes, which form a fairly uniform group (Tables 4 and 5). Apes have relatively untwisted heads, while humans have fourth MCs that have heads that are more pronated (Table 2). The Au. afarensis specimen is very pronated and most similar to humans while within the distribution of all taxa. The Swartkrans specimen (SKW 2954) is most similar to apes but within the distribution of humans (Fig. 3D). Although it has no evidence of a healed fracture, Susman (2004) suggested that this specimen, because of an uncharacteristically AP curved diaphysis and the presence of a ‘crook,’ may have been broken. If so, the torsion value for that specimen may be distorted and not reflect a normal morphology.

Finally, for the MC5, apes and humans have pronated heads (turned away from the thumb; Table 2) although humans have a statistically more twisted head than Pan, while all other taxa do not differ statistically (Tables 4 and 5). Australopithecus afarensis (n = 3) is variable, but on average, resemble Pan and Pongo the most, as does the one Swartkrans specimen (SKW 14147; Fig. 3E).

In base shape, humans are characterized by MC3 and MC4 that have pinched bases palmarly, while apes have bases that are relatively wider palmarly (Tables 6–8 and Fig. 4) Humans are statistically different from all taxa in MC3 base shape (Table 7). For the MC4, humans are statistically different from all apes except gorillas (Table 8), which have an MC4 base that is intermediate in shape between that of humans and chimpanzees. Australopithecus afarensis specimens (n = 5) are characterized by human-like, pinched MC3 bases, while Au. africanus (n = 2) and one specimen from Swartkrans are characterized by bases that are intermediate between that of apes and humans (while not being very different from three Au. afarensis specimens). The MC4 bases are more ape-like for Au. africanus and the Swartkrans specimens, while Au. afarensis is outside the variation of Pongo only, but falls closest to the median of humans.

Figure 4 Metacarpal base wedging.

Ratio of dorsal to palmar width of the base of MC3 and MC4. Higher ratios indicate a base that is more wedge-shaped, while a ratio of 1 indicates a base that is rectangular.

Table 6 Wedging values.

Dorsal to palmar medio-lateral width ratio of the third and fourth MCa.

Taxa	MC3	MC4	
H. sapiens (n = 29)	1.62	1.58	
	0.18	0.30	
P. troglodytes (n = 36)	1.32	1.20	
	0.13	0.15	
G. g. gorilla (n = 36)	1.35	1.42	
	0.11	0.21	
P. pygmaeus (n = 37)	1.27	1.09	
	0.14	0.10	
AL 333-16	1.55		
AL 333-65	1.53		
AL 333-153	1.56		
AL 333w-6	2.08		
AL 438-1	2.02		
AL 333-56		1.46	
Stw64	1.43		
Stw68	1.47		
Stw65		1.17	
Stw330		1.30	
SKX 3646	1.52		
SKX 2954		1.30	
Notes.

a For extant taxa, the mean is presented on the first line and standard deviation on the second.

Table 7 Extant taxa comparisons of MC3 wedging values.

Tamhane T2 post hoc comparisons of the dorsal to palmar medio-lateral width ratio for the MC3 (p-values, in bold when ≤0.05).

	H. sapiens	P. troglodytes	G. g. gorilla	
P. troglodytes	<0.001			
G. g. gorilla	<0.001	0.92		
P. pygmaeus	<0.001	0.57	0.08	

Table 8 Extant taxa comparisons of MC4 wedging values.

Tamhane T2 post hoc comparisons of the dorsal to palmar medio-lateral width ratio for the MC4 (p-values, in bold when ≤0.05).

	H. sapiens	P. troglodytes	G. g. gorilla	
P. troglodytes	<0.001			
G. g. gorilla	0.10	<0.001		
P. pygmaeus	<0.001	0.004	<0.001	

Discussion

The results for the first MC are as expected for humans with a head twisted toward the other fingers, probably in part to compensate for the reorientation of the trapezium in that species (Fig. 5; Lewis, 1977; Lewis, 1989; Sarmiento, 1994; Tocheri et al., 2005). As discussed above, the wider palmar aspect of the trapezoid, likely related to the palmar extension of its articulation with the capitate, results in a trapezium in the human hand that is pushed radially and rotated into alignment relative to the rest of the proximal carpal row (Lewis, 1977; Lewis, 1989; Sarmiento, 1994; Drapeau et al., 2005; Tocheri et al., 2005). This reorientation of the trapezium positions the MC1’s articular facet in a position that is more along the radioulnar axis of the other MC bases, in a position that is less advantageous for MC1 opposability. The strongly twisted head of the human MC1 reflects that species’ particular carpal morphology. The results for Pongo are intriguing given that it does not have developed thenar muscles (Tuttle, 1969) nor particularly large first MC articular surfaces on the trapezium (Tocheri et al., 2005). It is noteworthy that the strong inversion of the thumb and strong eversion of the second digit of Pongo (Fig. 6) is reminiscent of their value of metatarsal (MT) torsion (Drapeau & Harmon, 2013). A study of wild Bornean orangutans has shown that the hands and feet are more often used in grasps that involves the opposition of the pollex and hallux than in any other grips (including the hook grip and ‘double-lock’ grasp; McClure et al., 2012). This is particularly true of the hand where grips using the pollex in opposition were five times more common than grips using the lateral fingers only (McClure et al., 2012). Rearrangement of the muscles fibers to the distal phalanx of the pollex compensate for the absence or reduction of the tendon of m. flexor pollicis longus in Pongo (Tuttle & Cortright, 1988). The large torsion of the MC1 towards the palm is also surprising given that Pongo does not have a palmarly expanded trapezoid with a reoriented trapezium in the axis of more medial distal carpal row. The large degree of twisting is possibly needed to position the short pollex in opposition to the rigid palm of the hand instead of the much more mobile fingers. Their MC1-2 and MT1-2 morphology might reflect the importance of a strong opposing thumb-to-palmar and hallux-to-plantar surface grips in this highly arboreal taxon (Drapeau & Harmon, 2013). The torsion of the Australopithecus and Swartkrans MC1 specimens is similar to apes and probably reflects the lack of a human-like expansion of the palmar surface of the trapezoid and the lack of a human-like load distribution on the palmar surface (as suggested by Tocheri et al., 2008). The Swartkrans specimen (SK 84) is, of all the fossils, the specimen that most closely approaches the human form and falls within the range of distribution of humans. However, given its intermediate morphology, this study cannot resolve its taxonomical affinity (see Trinkaus & Long, 1990; Susman, 1994).

Figure 5 Metacarpal base and heads with average torsion values.

Metacarpal head (pale grey ovals) and base (dark grey quadrangles) of a left hand with the plantodorsal axes drawn (pale grey dotted line for the head; dark grey for the base; see methods for details). Metacarpal torsion is measured as the angle between these two axes in the coronal plane. The average torsion values are drawn from Table 2 and average wedging values of the MC3 and MC4 bases are drawn from Table 6. All drawings are aligned relative to the MC2-MC3 articulation. Relative orientation of the MC1 base (drawn for humans and chimpanzees only) is estimated from the orientation of the trapezio-MC articulation (from Fig. 20 in Sarmiento, 1994). Because of the strong wedging of the MC3 and MC4 bases, the dorso-palmar axis of the bases of the ulnar-most MCs of humans are more turned toward the thumb than in other taxa.

Figure 6 Patterns of metacarpal torsion.

Patterns of torsion for all MCs (median values for samples of n > 1).

For the MC2, there is no clear difference among species, extant or fossil. Previously observed torsion in humans relative to apes, as noted by Susman (1979) may have been an observation of the asymmetrical shape of the articular surface of the head. The lack of difference in torsion between dexterous humans and apes does not necessarily signify that the second finger of humans is used similarly to that of apes. In humans, depending on the grip used and the size of the object manipulated, the second finger may need to be either ulnarly or radially rotated. Unlike apes, humans are characterized by an asymmetrical MC2 head (Lewis, 1989), which allows the finger to axially rotate at the metacarpophalangeal joint. It is therefore possibly more advantageous to have a head that is only slightly twisted radially, which leaves flexibility to achieve different degrees of finger rotation for different types of grips. In addition, the human second MC, because of its morphology, might be capable of some axial rotation while that of apes is likely to be less mobile (Van Dam, 1934; Lewis, 1977; Lewis, 1989; Marzke, 1983; although El-shennawy et al., 2001, did not find significant rotation at that articulation in cadavers). Nonetheless, distal articular architecture in humans provides rotational flexibility of the finger necessary for a variety of effective grips. Interestingly, the base and head morphology of Australopithecus is clearly human-like (Marzke, 1983; Marzke, 1997; Marzke & Shackley, 1986; Drapeau et al., 2005; Tocheri et al., 2008; Kivell et al., 2011; Drapeau, 2012), which suggest human-like digit rotational capacities for these species.

The difference between humans and apes in torsion for the MC3 was expected and observed previously by Susman (1979). In apes, the torsion required to bring the head back into alignment with the other MC heads is minimal. In humans, finger supination is required in the power and precision grips (Landsmeer, 1955; Napier, 1956; Landsmeer, 1962; Marzke & Shackley, 1986). However, the third MC head is only slightly asymmetric compared to the second (Lewis, 1977; Lewis, 1989; Drapeau, 2012). As a consequence, the third MC head needs to be more supinated to allow for proper positioning of the finger during power and precision grips.

The relatively untwisted MC4 of African apes is not surprising. As for the third MC, these apes load that digit while knuckle walking (Inouye, 1994), which may favor a digit that flexes and extends closely to a parasagittal plane. Against expectations, the human MC4 is more pronated than that of apes. In humans, the fourth finger has an important role in buttressing (Susman, 1979). When buttressing, the fourth digit is flexed in the palm and ulnar torsion may help position the digit more appropriately. In the left hand, the predominant loading force may be the buttressing function rather than manipulation. Alternatively, it could be related to the degree of curvature of the metacarpal-base arch. Our measures of base wedging (Tables 6–8 and Fig. 4) have shown that humans have more palmarly wedged MC bases and therefore have a more arched MC base row than other extant large apes (Fig. 5; see also Reece, 2005). The dorsopalmar axis of the MC4 base is therefore more twisted towards the thumb in humans than in other large apes when in articulation with the other MCs and carpals (Reece, 2005). Because of the base orientation, the less radially twisted head of the MC4 in humans does not necessarily indicate that the whole digit is less radially twisted towards the thumb (Fig. 5). Further study of the orientation of the hand bones in vivo in apes will be needed to compare the actual degree of opposition of the MC and digits between humans and apes.

The lack of difference in torsion of the MC5 among humans, gorillas, and orangutans, which all have ulnarly twisted heads, also requires explanation. In humans, the articular surface of the MC5 head is also asymmetrical (Lewis, 1977; Lewis, 1989; Marzke, 1997), being somewhat a mirror image of the MC2. As a consequence, the digit is rotated towards the thumb during flexion, which is the natural position taken by the finger during power grips and some precision grips (Napier, 1956). Also, the MC itself is free to rotate slightly in humans though not in apes. These mechanisms may be sufficient during power grips and five-finger holds to produce a rotated digit without the need of the whole head to be twisted. Also, since the MC-base row is more arched than in apes (Reece, 2005; this study), the fifth MC base is rotated radially relative to the thumb (Fig. 5). Moderate ulnar torsion still leaves the fifth MC palmar surface in a radially facing position.

Australopithecus afarensis has MC3 and MC4 torsion values that are clearly more similar to humans, which suggest use of the hand in the fossil species that resembles humans more than apes. Similarly, their third and fourth MCs have wedge-shaped bases most like humans. The morphology of Au. africanus is less clearly similar to one species. Although torsion and base wedging values are within the range of humans, they are more typical of apes and their MC bases are not as wedged as humans. Together, these traits suggest that it may have been less adept at the pad-to-pad three-jaw chuck grasp relative to humans and A. afarensis, and may have been less adept at cupping the hand despite having relative thumb-to-finger lengths comparable to Au. afarensis (Green & Gordon, 2008; Rolian & Gordon, 2013). Evidence of some human-like loading in the trabecular patterns of the base of the MC1 and head of the MC3 (Skinner et al., 2015), combined with a weakened human-like signal in the trabeculae of the MC4 (Skinner et al., 2015; but see Almécija et al., 2015 for opposing view) concurs with this study’s observation that the ulnar side of the hand of Au. africanus is less human-like than that of Au. afarensis. Overall, the Au. afarensis morphology in torsion and base shape is human-like, while that of Au. africanus is less clearly human-like, suggesting that, perhaps, the medial palm was less suited to human-like manipulation than in Au. afarensis.

Torsion of the fifth MC, because it is not significantly different in humans, gorillas and orangutans is not particularly informative in Au. afarensis. The morphology of the base in that species suggests less mobility in flexion and supination at that joint than in humans (Marzke, 1983; Marzke & Shackley, 1986; Marzke, Wullstein & Viegas, 1992). However, as for the second MC, the MC5 head is asymmetric (Bush et al., 1982; Marzke, 1997; Drapeau, 2012; Ward et al., 2012). This mosaic of ape and human traits in the fossils species indicates an intermediate state, in which the human-like involvement of the fifth finger in manipulation might be limited to the phalangeal segment of the digit and to a more radially turned hypothenar region. In addition, the Au. afarensis hands did not have a robust thumb nor a styloid process on the MC3 (Bush et al., 1982; Marzke, 1983; Drapeau et al., 2005; Ward et al., 2012) which indicates that these taxa were not incurring as large loads on the thumb and on the palm of the hand. These traits are more human-like in the Au. africanus specimens (Ricklan, 1987; Kivell et al., 2011; Supplemental Information), suggesting adaptations to greater loads in the lateral hand of that later taxa. The differences between the two fossil taxa are not large, but they might indicate slightly different adaptations to manipulation possibly reflecting slightly different evolutionary paths.

The curved MC base arch of humans orients the ulnar MC bases with their palmar surface toward the thumb. As a consequence, when the fifth and, to a lesser degree, fourth MC are flexed in humans, it produces the typically human cupping of the palm that is used in power grips of large objects (Peters & Koebke, 1990). The greater arching of the MC bases might then be an adaptation of such movement in humans and pronation of the MC4 head is only a consequence of the reoriented base. If so, this would indicate that Au. afarensis, with its wedged bases, has begun the reorientation of the medial aspect of the palm of the hand despite probably not being able to flex the MC4 and MC5 as much as humans (Marzke, Wullstein & Viegas, 1992). Combined with the asymmetry of the fifth MC head, Au. afarensis was probably capable of a power and cradle grips that were not completely human but approached it significantly.

The torsion of Swartkrans MCs can be characterized, as a whole, as being more ape-like than human-like. However, of all the three Swartkrans specimens available for analysis, the MC4 is the only one that is more clearly ape-like (although still within the range of humans) by being more radially twisted. This morphology is rather surprising considering that the MC3 base tends to be pinched suggesting a fairly deeply arched MC-base row. These conflicting results tend to support Susman’s (2004) interpretation that SKW 2954 was fractured and is likely to be pathological and distorted. As a whole, the Swartkrans specimens are not particularly informative with respect to manipulative dexterity, although some traits, such as moderate base wedging, does point toward some adaptations for that behavior.

It is unknown whether metacarpal torsion is genetically determined or whether it is plastic, or a combination of both, but variation in metatarsal torsion among human populations with various types of footwear (Drapeau & Harmon, 2013; Forgues-Marceau, 2013) as well as variation in humeral torsion according to throwing activity (e.g., Pieper, 1998) suggest that it is a trait that is at least in part plastic in the foot, arm and possibly in the hand. If so, this trait would be particularly informative on the actual use of the hand (Lovejoy, Cohn & White, 1999; Ward, 2002), but further work is needed on variation in metacarpal torsion and how it may be a plastic response to specific loading regiments. Irrespective of whether metacarpal torsion is completely, partly or not at all determined genetically, when studies are combined with metacarpal base shape (which is much more likely to be genetically determined), it is informative on the use of the hand and reflects, the capacity to do a three-finger chuck and to cup the palm of the hand.

Discussions of hand evolution often assumed that the human hand evolved from a form similar to that of African great apes. However, recent work has shown that the ape hand, particularly that of chimpanzees, might be derived relative to that of the Pan-Homo last common ancestor (Drapeau et al., 2005; Drapeau & Ward, 2007; Lovejoy et al., 2009; Almécija, Smaers & Jungers, 2015). More specifically, there is growing evidence that the long hands of Pan are derived (Drapeau et al., 2005; Drapeau & Ward, 2007; Almécija, Smaers & Jungers, 2015) and that the thumb to digit ratio of humans and gorillas is closest to that of the primitive form for hominins (Almécija, Smaers & Jungers, 2015). The discovery of a nearly complete Ar. ramidus hand, which is characterized by a continuous capitate-MC2 articular facet, and a mobile hamate-MC5 joint, has led Lovejoy and colleagues (2009) to argue that these traits, because they were present in the early Miocene Proconsul, are primitive for hominins. This interpretation implies that all extant apes stiffened their hands at the carpometacarpal joints independently. A closer look at the morphology of the mid- and late Miocene apes reveals that those for which this morphology is known are all characterized, without exception, by a planar, discontinuous capitate-MC2 joint (Sivapithecus, Rose, 1984; Rudapithecus hungaricus, Kivell & Begun, 2009; Oreopithecus bambolii, M Drapeau, pers. obs., 2000; Hispanopithecus laietanus, S Almécija, pers. comm., 2015; Pierolapithecus catalanicus, S Almécija, pers. comm., 2015) including in Pierolapithecus, which displays no obvious adaptations to suspensory behavior (Moyà-Solà et al., 2004) and Sivapithecus, which probably was still pronograde (Pilbeam et al., 1990). It is more parsimonious from all the available evidence to assume that the last common Pan-Homo ancestor had a discontinuous facet and therefore a lateral palm that was rigid. The non-continuous facet on the Au. anamensis capitate (Leakey et al., 1998; Ward, Walker & Leakey, 1999) is intriguing given that the older Ar. ramidus had a continuous facet. Either its poor preservation obscures a continuous facet on the Au. anamensis capitate or Ar. ramidus is autapomorphic and convergent on Au. afarensis for that trait. More specimens of Au. anamensis are needed to resolve this issue.

The hamate-MC5 joints of Miocene hominoids, when known, have a joint surface morphology that is not believed to have allowed much motion comparable to that of extent apes (Sivapithecus parvada, Spoor, Sondaar & Hussain, 1991 or have an articular surface that extend on the hamulus (Hispanopithecus and Pierolapithecus, S Almécija, pers. comm., 2015), although the hamulus in these taxa is not as proximally projecting as in extant apes. However, the fossil taxa probably had a more mobile hamate-MC5 joint in dorsiflexion as suggested by an articular surface that extends to the dorsum of the MC base (Pierolapithecus; Almécija et al., 2007; S Almécija, pers. comm., 2015) or by a similarity to taxa that dorsiflex (Proconsul; Napier & Davis, 1959; O’Connor, 1975). If the morphology is interpreted accurately, it implies that the last Pan-Homo common ancestor was characterized by a somewhat mobile joint, a morphology also seen in Ar. ramidus (Lovejoy et al., 2009).

The Miocene fossil evidence suggests that the Pan-Homo last common ancestor had thumb to digit proportions that were close to that of humans, a rigid, planar bipartite capitate-MC2 joint, and possibly a moderately mobile hamate-MC5 joint. The morphology of Ar. ramidus, a likely ancestor to Australopithecus (White et al., 2009), indicates that a hamate-MC5 joint capable of plantarflexion and a continuous capitate-MC2 joint (but that was not curved as in humans and Au. afarensis) had already been transformed by 4.4 Ma (Lovejoy et al., 2009). Almécija and colleagues even propose that pad-to-pad grips were possible by 6 Ma with Orrorin tugenensis (Almécija, Moyà-Solà & Alba, 2010). These traits associated to more dexterous manipulation in hominins have appeared long before any evidence of stone tools (Panger et al., 2002; Almécija, Moyà-Solà & Alba, 2010; Drapeau, 2012; Almécija & Alba, 2014; Almécija, Smaers & Jungers, 2015) and is unlikely to be an adaptation to that specific behavior. It probably reflects adaptation to increase dexterity in the context of habitual bipedality and a relaxed selection for locomotor adaptation of the upper limbs (Almécija, Moyà-Solà & Alba, 2010; Drapeau, 2012). Further transformations in Au. afarensis, such as a capitate-MC2 surface that is more curved and oriented in a more transverse plane, MC head asymmetry and, possibly, radio-ulnar arching of the MC base row may be a response to increased reliance on precise and forceful grips required for stone tool use (McPherron et al., 2010) and possibly even stone-tool manufacture (Harmand et al., 2015) in that taxon. Further adaptation, such as a more robust thumb, a styloid process and a palmarly expanded trapezoid appear later in time, possibly only in Homo (Berger et al., 2015), and testify to the continued importance of manipulation in the evolution of our lineage.

Conclusions

Metacarpal head torsion is different between humans and apes, particularly in the third and fourth MCs. For the MC2 and MC5, articular morphology, including head asymmetry, may be a better indicator of human-like manipulation and rotational capacity of the digits. Differences in head torsion among species are broadly as expected, except for the fourth and fifth MCs which are generally less radially twisted in humans. These unexpected results for the ulnar part of the hand might relate to how the MC bases are positioned relative to each other and to the degree of curvature of the proximal metacarpal arch, a curvature that is greater in humans than in apes due to greater base wedging of the third and fourth MCs.

An overall view of the Au. afarensis and Au. africanus MCs is consistent with previous analyses of the hand in these species. The lack of ulnar twist in the pollical MC suggest that these species were probably not characterized by a palmarly expanded trapezoid that positioned the trapezium in line with the rest of the carpals and, according to Lewis (1977; Lewis, 1989; Tocheri et al., 2005), allowed for compressive loads from the base of the first MC to dissipate through the palmar aspect of the palm via a palmarly expanded trapezoid and palmar trapeziocapitate articular facet. These fossil species likely had a primitive configuration similar to apes with a trapezium positioned more perpendicular to the rest of the distal carpal row and therefore were not able to dissipate compressive loads from the thumb through the palm as effectively as modern humans. Previous studies had shown that the second MC of Australopithecus was modified from the assumed primitive morphology, with a base and head allowing for some movement of the digit, but the third lacked the human-like styloid process, suggesting only a partial transition towards a human-like grip. This study has shown that Au. afarensis had human-like orientation of the third and fourth MCs, indicating the possibility of adequate three- or four-jaw chucks in these species (although possibly with less ulnar deviation of the thumb than in humans; Marzke, 1992). More ulnarly, there is less evidence of a human-like grip, but for the asymmetry of the fifth MC head that allows for phalangeal axial rotation, suggesting that active involvement of the fifth digit in a five-jaw chuck was probably limited to the phalanges. However, the shape of the MC3 and MC4 bases suggest a configuration of the MC base row that was more arched and human-like in Au. afarensis, allowing for more opposition of the fifth MC than is possible in large apes. As a consequence, although Au. afarensis had not developed a completely human grip, it showed significant derived traits that suggest that there was directional selection for improved dexterity and strength in various grips in these early hominins, adaptations that appear to have begun with Ar. ramidus (Lovejoy et al., 2009) and even possibly in Orrorin tugenensis (Almécija, Moyà-Solà & Alba, 2010). In contrast, Au. africanus is less clearly human-like than Au. afarensis since it presents MC3 torsion more typical of apes, has an MC3 base shape that is more intermediate between apes and humans, and an MC4 base that is more ape-like. This suggests that Au. africanus may have been less dexterous in the three-jaw chuck and cradle grips than Au. afarensis despite thumb-to-fingers proportions that were probably similar (Green & Gordon, 2008; Rolian & Gordon, 2013). Overall, this study supports previous studies on Au. afarensis and Au. africanus that these taxa had derived hand morphology that suggest increase finesse and strength in pad-to-pad, two- and three-jaw chucks grips and some adaptation to human-like power grips and support the hypothesis that human-like manipulation largely predated Homo.

Supplemental Information

Supplemental Information 1 Metacarpal torsion data

Click here for additional data file.

Supplemental Information 2 Metacarpal (MC3 and MC4) wedging data.

Click here for additional data file.

The author would like to thank Mamitu Yilma and Alemu Admessu from the National Museum of Ethiopia; Drs. William H. Kimbel and Donald Johanson, Institute of Human Origins, Arizona State University; Dr. Yohannes Haile-Salessie and Lyman Jellema, Cleveland Museum of Natural History; Dr. Jerome Cybulski and Dr. Janet Young, Canadian Museum of Civilization; Nunavut Inuit Heritage Trust; Dr. Richard W. Thorington and Linda Gordon, National Museum of Natural History.

Additional Information and Declarations

Competing Interests

Author Contributions

The author declares there are no competing interests.

Michelle S.M. Drapeau conceived and designed the experiments, performed the experiments, analyzed the data, contributed reagents/materials/analysis tools, wrote the paper, prepared figures and/or tables, reviewed drafts of the paper.

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
