# Peer review of "Metacarpal torsion in apes, humans, and early Australopithecus: implications for manipulatory abilities"

_PeerJ, doi:10.7717/peerj.1311_

## Round 0.1 · original submission · Major Revisions

Dear Michelle,

I now have two reviews for your paper. Each notes strengths of the methodology and agree that this analysis and the data included are worthy of publication. However, both reviewers express concerns regarding the particular interpretations generated from the data. One argues (Sergio Almécija) for minor changes given that the differences among species and the functional interpretations may be less clear-cut than described. Another recommends a major revision expressing specific concerns regarding inferences of the condition of the LCA and the developmental origin of the character in questions (i.e., developmental patterning or remodeling in response to functional loading). He also suggests addressing the some additional fossil specimens. These issues can impact the conclusions drawn. As these issues are not fundamental to the methodology and largely interpretive they are quite likely surmountable with revision. However, as these include suggestions beyond simple editorial corrections I have classified this as a “Major Revision”. If you agree that you can address these concerns, I would look forward to your revised manuscript. Each reviewer also provides a number of specific comments that you will find helpful.

If you have any questions please do not hesitate to ask.

Best,

Phil Reno

Reviewer 1 ·

Basic reporting

43 typo – ‘manipulatory’
45 Here, should note orientation of medial MC torsion, i.e., towards the pollical side of the hand.
45-46 Should sentence read, ‘Torsion of the second and third MC head in humans and the apes is a trait that has been described qualitatively, but has not been extensively quantified (except in humans; Singh, 1979; Peters and Koebke, 1990) . . .’
97,101,103 check grammar
117 Stating that the australopithecines acquired these characteristics of the metacarpal heads suggests that they are absent in earlier hominins. Should the author comment on the hand of Ar. ramidus?
167 omit ‘all’
172 ‘based’
273 ‘less stabilizing selection’ This implies that variance in the torsion of the MCs is determined solely by genetic coding and that an individual’s fitness depends on torsion. Is it possible that some variance may be due to cartilage modeling?
274-281 This discussion seems contradictory. In Pongo, this MC1-2 orientation is considered to ‘reflect the importance of a strong opposing thumb-to-palm grip’. Later, the australopiths and SK specimen are ape-like reflecting a lack of human-like load distribution. Don’t humans also have strong ‘opposing thumb-to-palm grip’? How can Pongo with a reduced thenar musculature and absence of a flexor pollicis longus tendon have a strong thumb-to-palm grip?
296 the amount of axial rotation at CMC2 in humans is perhaps overstated.
304-11 That Pongo has a similar degree of rotation as the African apes suggests that this is not due to knuckle-walking.
309 Again, the amount of axial rotation at the CMC2 is overstated. The orthopaedic literature indicates little rotation or motion at that joint. Any rotation that would occur at this joint would be due to imposed external forces and not the result of muscular activity.
326-8 Data are available on the in vivo orientation of the hand bones.
359+ “this sequence . . . but occurred earlier in a form ancestral to A. afarensis, a hypothesis that can be tested with the discovery of slightly older fossils that would preserve the MC2 head.’ While neither ramidus nor anamensis has a MC2 head, at least mention of these taxa should be made.
395+ The early hominins are considered primitive relative to modern humans with a ‘primitive condition similar to apes’. However, according to the data, Pongo has a highly twisted pollical MC. Does this affect the interpretation of torsion and function?
397 The mechanism by which load dissipation is achieved must be discussed. Also, what is the significance of load dissipation in the lateral carpus?
403-05 This statement implies that afarensis may be the first hominin to achieve ‘human-like orientation of the third and fourth fingers’ and the primitive condition is exemplified by the African apes. This was falsified by the publication of Ar. ramidus.
407 should read ‘allows for’
407-408 Author should recognize that there is the capacity for rotation of MC5 at the CMC5 joint so not all motion occurs at the MCP joint. Observation of the afarensis MC5 base and the presence of a ridge for the opponens digiti minimi muscle indicates there was some axial rotation at the CMC5 joint.
411-13 ‘it showed significantly derived traits that . . .’. Derived from what? This implies that the apes retain the primitive condition. Again, reference to ramidus would help clarify this.
426+ As above, reference to ramidus is needed here.
428 note that Green & Gordon come to a different conclusion (not stone tool producing hominins) based on their analysis of the africanus metacarpals.
507 typo
541 typo
Figure 2 Does use of these figures require permission from authors/publishers? The modifications seem minor.
Figure 5: Legend should identify the various lines (e.g., dark dotted lines are the basal axis, light dotted lines indicated the head’s axis). Also, should note that these are mean values and should redirect to table 2 for full descriptive stats.
Figure 6: This is a restatement of Fig. 3 and seems redundant. If the figure is kept, please include measures of variance in the species values in addition to mean. The connecting lines can be omitted.
Table 3: It appears that angles are side specific. For example, do the MC5s AL 333-141 and SKW14147 have the same degree of rotation and that the difference in sign is due to one is from the right side and the other is left? See also the right and left antimeres from AL 468-1. This is confusing and the author should choose a standard (e.g., if it is pronated, it should be +, if supinated -) as described on lines 193-196, 213-214.
Is SKW2954 a right or a left?
Table 5: Author should note that values above the diagonal are the absolute difference in angle between the pair-wise comparisons.
Table 6: Pongo sample size is greater here than listed in Table 1. Perhaps Tab 1 should clarify that it is the sample for the torsional analysis only.

Experimental design

The hand looms large in our identification of the suite of anatomies and behaviors that are central to our interpretation of key hominin adaptations. The core of this analysis is to identify those characters – with an emphasis on metacarpal torsion – that post-3.5 Ma hominins share and distinguish them from the extant African apes. The implication is that we can assess the quality of hand function in the early hominins vis-à-vis the African apes. The main conclusion is that Au. afarensis & africanus are derived in their hand function from an ape-like primitive condition allowing greater amounts of human-like dexterous activities, notably stone tool use and manufacture.
This position implies that the African apes retain the primitive anatomy and that the hominin hand is highly derived. With the recovery of the Ar. ramidus fossils, it is clear that the hand of the apes is very highly derived in ways that reduce fine motor ability and that the ramidus hand is a generalized manipulative organ. The model of analysis implied here (apes primitive – humans derived) is no longer tenable and the recently published papers should be consulted and referenced. That there is no mention of the nearly complete ramidus hand at all is a surprising omission and needs to be addressed (Lovejoy et al 2009 Science [forelimb, Last Common Ancestor]). Also, there is no mention of Au. anamensis. While there are no metacarpals known published for the species, the presence of a capitate can inform this discussion (Ward, et al 2001 JHE). No mention of Au. sediba even though Kivell et al is cited? Another surprising citation omission is Ward et al 2012 (JHE) summary of the afarensis postcrania. No citations by Ricklan who studied the africanus hand?
What is equally surprising is that there is not mention about the absolute lengths of MC2-4. This feature is absolutely essential in understanding the nature of the specialized hand of the apes. The only mention in the paper about length is of the relative length of the pollex. Any discussion of function of the hand should include this observation. In addition, that the apes have also lost significant thenar muscle mass and the Flexor Pollicis Longus tendon should be included in the discussion since it impacts the functional capacity of the thumb as well as its reduced role in manipulative behaviors. These are specializations and not primitive retentions.
Evidence exists that tool making/using predates 3 Ma. There is also evidence that chimps can and do use stone tools. By the logic of this paper, the metacarpal torsion observed in the apes is inconsistent with their ability to use tools. Ramidus, with metacarpal and phalangeal proportions not dissimilar from afarensis then should be a tool-maker/user although there is no evidence for tool use at that time. Also, intensive survey of the Hadar deposits over the past 40+ years has not yielded a single stone tool. The finds at nearby Dikika – while compelling – require independent verification of additional evidence. Perhaps a simpler argument is that the early hominins retained the metacarpal proportions and overall anatomy of the LCA and that the ape hands are highly derived. Thus, the anatomy observed here may have little to do with tool use/manufacture.

Validity of the findings

The observations appear reliable and useful.

Additional comments

The data are a valuable contribution to the field. However, a couple issues prevent publication at this time. The basic model that the apes retain the primitive condition and hominins derived from that condition has been falsified by the recovery of older hominin fossils. Describing ways in which humans and apes differ in the details of their hands says more about the specializations in the ape hands more than it does about the hominin hand. The complete absence of mention of key and relevant taxa and omission of significant publications must be addressed. I opted for revise and resubmit rather than reject since I believe that these are solvable problems and that the data need to see the light of day.

·

Basic reporting

In this work, Drapeau presents a straightforward analysis of metacarpal (I-V) torsion (and base wedging in mc3 and mc4) in extant hominids, Australopithecus and available metacarpals from Swartkrans. The paper is clearly written and well presented, and the results show clear trends in the torsion of some human metacarpals (as compared to great apes). Humans are statistically different from African apes for mc1 and from the three great ape genera for mc3 and mc4.

Experimental design

No comments.

Validity of the findings

There is not a single clear cut result in any of the analyses (there is extensive overlap in the range of humans and great apes). This is problematic because, in most cases, the fossils fall within the overlapping regions of humans and great apes, which makes it difficult to draw specific inferences about affinities of these fossil taxa (see my "General Comments for the Author" section). In any case, these results on metacarpal torsion are new and of great interest.

Additional comments

I would love to see this paper published soon, but I would suggest to tone down some statements about A. afarensis being clearly more human-like than others (this might be the case, but it is not fully supported in this study). For example, the idea that A. africanus is more ape-like than A. afarensis in terms of mc torsion: this is only supported by the mc3, which is still within the human range of variation. Similarly, the author states that (pages 16-17): "The torsion of Swartkrans MCs can be characterized, as a whole, as being more ape-like than 
human-like. However, of all the three Swartkrans specimens available for analysis, the MC4 is 
the only one that is clearly ape-like by being more radially twisted." Although I am aware of the "trends" in mc torsion reflected in Figure 3, for the sake of fairness, I insist that the author should recognize again that the Swartkrans metacarpals all fall within the human range too.

The evolution of the hand in apes and humans is one of my favorite topics, and I agree that the overall hand morphology of Australopithecus was more human-like than ape-like, and therefore these guys were capable of advanced manipulative tasks. This is a complex topic and I think that the present discussion/conclusions about hand proportions and hand use in early hominins would be enriched if considering also other ideas discussed in some papers of my own: Almécija & Alba 2015 and Almécija et al., 2015 are critics to Rolian and Gordon 2013 and Skinner et al., 2015 respectively. Almécija et al., 2010 specifically discusses the origins of precision grasping in hominins (which probably appeared before the advent of widespread stone tool making).

Other minor comments are:
- When describing the results on torsion, and which groups are "different" from humans, I would recommend that the author specifies that they are "statistically different" (since there is always overlap in the ranges) and provide the p-value.
- The use of "australopithecines" implies a monophyletic subfamily (i.e., Australopithecinae). This is far from settled (e.g., is Paranthropus sister of Australopithecus, or sister of Homo?). I would recommend using the vernacular term "australopith" to circumvent the problem.
- page 3: "Apes are much less dexterous than humans in manipulation (Napier, 1960) and have much more difficulty in pad-to-pad grips". I would add here Christel 1993 (in Hands of primates). This author uses a different nomenclature for "pad-to-pad", but in this work it is clearly stated that, amongst modern hominoids, this grip is only observed in humans.
- page 3: "Apes are much less dexterous than humans in manipulation (Napier, 1960) and have much more 
difficulty in pad-to-pad grips. This is in part a consequence of their relatively long fingers and 
short thumbs." These statements should be backed up also with other relevant bibliography on the matter: e.g., Schultz, 1930; Ashley-Montagu, 1931; Midlo, 1934; Jouffroy et al., 1991; Christel, 1993; Watkins et al., 1993; Susman, 1994.
- page 3: "All great apes are characterized by a hook grip, which involves 
flexing all the fingers in sagittal planes (Lewis, 1977, 1989; Napier, 1993), a position that is 
remains the same when using the hand in terrestrial locomotion." I find this last assertion to be a bit confusing. It might be the case for orangs during fist walking, but knuckle walking requires full extension of (at least some) metacarpophalangeal joints. Also, delete the "is"?
- page 12, line 277: "taxon" instead of "taxa"?
- page 13, starting line 306: talking about mc3..."Among apes, knuckle-walkers have the less twisted heads, 
which may reflect the habitual use of the hand in terrestrial locomotion, which loads heavily the third digit (Inouye, 1994)." The first statement is not supported by any data in this work. Please, reword accordingly. The remaining paragraph on human grips should be referenced.
- page 15, starting in line 353: "the morphology of A. afarensis fifth MC suggests, on average a morphology that is more Pan-like, with untwisted heads." Figure 3E shows that the three A. afarensis inspected fall within the ranges of all the great apes and humans. What is this statement based on?

---

## Round 0.2 · Minor Revisions

Hi Michelle,

Thank you for providing your revised manuscript. You have done a thorough job responding to the reviewers’ comments and a very nice manuscript has been the result. I have provided some minor editorial comments below. I have classified this as Minor Revisions as opposed to Accept, as this submitted version is send directly to the production phase.

Line numbers refer to the reviewing pdf.

212 – to the other AS a function of

226 – taxON

301 – aN MC4

335 – I suggests: thumb-to-palmer or hallux-to-plantar surface grip

354 – Add a comma after morphology

358 – a VARIETY OF effective grips

397 – Moderate ulnar torsion

412 – This seems unclear. I suggest (if this is what you intend) “that the ulnar side of the hand of Au. africanus is less human-like than that of Au. afarensis.”

451 – informative WITH RESPECT TO manipulative

462 – when studies ARE combined

467 – particularLY

493 – Either: MC5 jointS or HAS

562 – has aN MC3

563 – and aN MC4

Table 2: I suggest including the anatomical significance of positive and negative values as described in the methods to Footnote 2 as well.

---

## Round 0.3 · accepted · Accept

Hi Michelle,

Thank you for your quick response. This paper provides important data for understanding the evolution of manipulative abilities in hominids. While debates regarding ancestral conditions will continue, I'm sure, this paper is a great addition.